# Acute Stress in Health Workers during Two Consecutive Epidemic Waves of COVID-19

**DOI:** 10.3390/ijerph19010206

**Published:** 2021-12-25

**Authors:** Kathrine Jáuregui Renaud, Davis Cooper-Bribiesca, Elizabet Martínez-Pichardo, José A. Miguel Puga, Dulce M. Rascón-Martínez, Luis A. Sánchez Hurtado, Tania Colin Martínez, Eliseo Espinosa-Poblano, Juan Carlos Anda-Garay, Jorge I. González Diaz, Etzel Cardeña, Francisco Avelar Garnica

**Affiliations:** 1Unidad de Investigación Médica en Otoneurología, Instituto Mexicano del Seguro Social, Ciudad de Mexico 06720, Mexico; adan.miguel@imss.gob.mx; 2Departamento de Psiquiatría, Hospital de Especialidades del Centro Médico Nacional Siglo XXI, Instituto Mexicano del Seguro Social, Ciudad de Mexico 06720, Mexico; coop_2000@yahoo.com (D.C.-B.); elizabetmtz27@gmail.com (E.M.-P.); 3Departamento de Anestesiología, Hospital de Especialidades del Centro Médico Nacional Siglo XXI, Instituto Mexicano del Seguro Social, Ciudad de Mexico 06720, Mexico; drarascon@hotmail.com; 4Departamento de Terapia Intensiva, Hospital de Especialidades del Centro Médico Nacional Siglo XXI, Instituto Mexicano del Seguro Social, Ciudad de Mexico 06720, Mexico; lashmd@gmail.com; 5Departamento de Admisión Continua, Hospital de Especialidades del Centro Médico Nacional Siglo XXI, Instituto Mexicano del Seguro Social, Ciudad de Mexico 06720, Mexico; tania.colin@imss.gob.mx; 6Departamento de Inhaloterapia y Neumología, Hospital de Especialidades del Centro Médico Nacional Siglo XXI, Instituto Mexicano del Seguro Social, Ciudad de Mexico 06720, Mexico; gadaca1962@yahoo.com.mx; 7Departamento de Medicina Interna, Hospital de Especialidades del Centro Médico Nacional Siglo XXI, Instituto Mexicano del Seguro Social, Ciudad de Mexico 06720, Mexico; estumed@hotmail.com; 8Departamento de Imagenología, Hospital de Especialidades del Centro Médico Nacional Siglo XXI, Instituto Mexicano del Seguro Social, Ciudad de Mexico 06720, Mexico; jorge.gonzalezd@imss.gob.mx (J.I.G.D.); francisco.avelar@imss.gob.mx (F.A.G.); 9Center for Research on Consciousness and Anomalous Psychology, Department of Psychology, Lund University, 22100 Lund, Sweden; etzel.cardena@psy.lu.se

**Keywords:** COVID-19, health workers, stress, anxiety, depression, sleep

## Abstract

The COVID-19 pandemic has provoked generalized uncertainty around the world, with health workers experiencing anxiety, depression, burnout, insomnia, and stress. Although the effects of the pandemic on mental health may change as it evolves, the majority of reports have been web-based, cross-sectional studies. We performed a study assessing acute stress in frontline health workers during two consecutive epidemic waves. After screening for trait anxiety/depression and dissociative experiences, we evaluated changes in acute stress, considering resilience, state anxiety, burnout, depersonalization/derealization symptoms, and quality of sleep as cofactors. During the first epidemic wave (April 2020), health workers reported acute stress related to COVID-19, which was related to state anxiety. After the first epidemic wave, acute stress decreased, with no increase during the second epidemic wave (December 2020), and further decreased when vaccination started. During the follow-up (April 2020 to February 2021), the acute stress score was related to bad quality of sleep. However, acute stress, state anxiety, and burnout were all related to trait anxiety/depression, while the resilience score was invariant through time. Overall, the results emphasize the relevance of mental health screening before, during, and after an epidemic wave of infections, in order to enable coping during successive sanitary crises.

## 1. Introduction

Stress can be considered as an expression of alarm towards stimuli arising from the environment or from internal cues that are interpreted as a disruption of homeostasis, including physiological and behavioral responses [1,2]. Several brain areas participate in modulating stress responses through the sympathetic nervous system, and the hypothalamic–hypophyseal–adrenal axis, with individual variability related to both genetic differences and life experiences (for a review, see [3]). However, in healthy adults, anxiety or depressive symptoms may alter hypothalamic–hypophyseal–adrenal axis functioning and contribute to allostatic load and disease [4]. Conversely, the ability to adapt stress responses to a challenge (or resilience) may allow for recovery from stressors [5].

Behavioral/experiential responses to stress may include altered perceptions of the self or the environment (dissociative experiences), as well as intrusive, avoidance, and arousal symptoms that can be conceptualized as attempts to adapt to the stressful situation, limiting painful thoughts and feelings associated with the event, and allowing the person to function [6]. However, stress reactions may also modify performance, verbal memory, assessment of risk, and decision making [7,8] and interact with impulsivity [9]. In health workers, these effects may be detrimental for the quality of healthcare and patient safety [7,10].

Stress responses are tuned to environmental uncertainty [11], and the characteristics of the COVID-19 pandemic have provoked generalized uncertainty [12,13]. Globally, the pandemic has been increasingly associated with mental and neurological manifestations, and it is likely to exacerbate pre-existing mental health, neurological, and substance use disorders, in both health workers and the general population [14]. The World Health Organization has been working to promote the integration of mental health and psychosocial support into the COVID-19 response; the strategies may guide efforts to strengthen mental healthcare during recovery, as well as preparedness before the next emergency [14].

During the pandemic, health workers have been experiencing psychological symptoms and mental health problems, including anxiety, depression, burnout, insomnia, and stress [15,16,17,18]. In the first epidemic wave, in the United States of America, more than half of 1399 health workers completing a national survey on mental health endorsed at least mild psychiatric symptoms, and approximately forty percent endorsed symptoms suggesting a clinically significant emotional disorder [19]. A systematic review of 117 studies including 119,189 participants showed a pooled prevalence of 40% for acute stress, 30% for anxiety, 24% for depression, and 28% for burnout [20]. A cross-sectional study in Argentina showed hair cortisol values outside of the healthy reference range in 40% of 234 health workers, which was related to their perceived stress [21].

Health workers are known to experience high levels of distress related to their occupation demands [22]. During the COVID-19 pandemic, they have been exposed to a variety of situations outside their ordinary experience that imply increased working demands, alongside scarcity of resources, high mortality rates, and ethical issues, with cumulative stress [23,24]. Additional sources of stress in the workplace include [25,26,27,28]: inadequate supplies of personal protective equipment; heightened risk of SARS-CoV-2 infection and contagion to relatives; uncertainty of care for family needs if infected or away; not having rapid access to testing through occupational health if required; overwhelming workload; and not being able to provide competent care if deployed to a new area.

The psychological responses of health workers to a crisis are multi-factorial, while short-term and long-term stress may result in different regulatory mechanisms, whose consequences can modify the response to successive events (for reviews, see [3,29]). Although the effects of the pandemic on the mental health of health workers may change as the pandemic evolves, the majority of reports have been cross-sectional, web-based studies performed during the first half of 2020 (for reviews, see [18,30,31]). Information on the psychological reactions of frontline health workers related to repetitive stressors, within the same working conditions, can contribute to planning future strategies to address new health crises.

In a hospital reconfigured to address the surge in patients with COVID-19, we conducted a study to assess acute stress in frontline health workers during the first two consecutive epidemic waves of COVID-19 (from April 2020 to February 2021) while monitoring quality of sleep, depersonalization/derealization symptoms, burnout, state anxiety, and resilience. The study group was evaluated five times, before, during, and after two consecutive epidemic waves. Since, after each evaluation, we provided individualized reports of the results to the corresponding clinical department, to take into account the possible effect of being observed during the first epidemic wave, we also evaluated a second group of frontline health workers just during and after the second epidemic wave; during the first wave, they worked in the same hospital conditions, but did not participate in the research.

## 2. Materials and Methods

The local institutional research and ethics committees approved the research protocol (R2020-3602-042 with amendment), and all the participants gave their written informed consent to participate in the study.

### 2.1. Participants

A total of 138 health workers (33.7 ± 8.0 years old, 82 women and 56 men) at the forefront of patient care participated in the study, after an open invitation, with no sample size calculation. According to the time of inclusion in the study, either the first or the second epidemic waves, participants were categorized into two groups:Group I: A total of 109 frontline health workers (33.6 ± 7.9 years old, 60 women and 49 men), among 204 candidates who had previously completed psychological evaluations during the first epidemic wave in order to assess posttraumatic stress disorder [32] and kept their position in the same hospital.Group II: A total of 29 frontline health workers (34.0 ± 8.7 years old, 22 women and 7 men), who accepted to participate in the study during the second epidemic wave, but were not invited to participate in the previous study (in the first wave). They also kept their position in the same hospital during the two epidemic waves.

Additionally, 22 (13.7%) workers accepted to participate but did not complete the study protocol. They were 22 to 53 years old (11 women and 11 men). One of them was absent due to sick leave, and twenty-one dropped out from the study at the last evaluation; among them, 17 (13.4%) participated during the first epidemic wave (Group I), and 5 (14.7%) accepted to participate during the second wave (Group II) (Appendix A Figure 1).

### 2.2. Procedures

During follow-up, the time points of evaluation were (Figure 1):

Group I.

While the clinical departments were reconfigured for COVID-19 and working teams were reorganized (7 to 16 April 2020);During the peak of inpatient admissions of the first epidemic wave (30 April to 24 May 2020);Before clinical departments were re-opened for patients with diseases other than COVID-19 (17 to 30 July 2020);During the peak of inpatient admissions of the second epidemic wave (7 to 28 December 2020);Before clinical departments were re-opened for patients with diseases other than COVID-19, and during COVID-19 vaccination for health workers (26 January to 19 February 2021).

Group II.

During the peak of inpatient admissions of the second epidemic wave (7 to 28 December 2020) (evaluation 4 in Group I);Before clinical departments were re-opened for patients with diseases other than COVID-19, and during COVID-19 vaccination for health workers (26 January to 19 February 2021) (evaluation 5 in Group I).

At the time of inclusion in the study (evaluation 1 in Group I, and evaluation 4 in Group II), after documenting demographics and a short medical history, psychological screening was performed using three self-administered inventories (Section B.1): the Hospital Anxiety and Depression Scale by Zigmond and Snaith 1983 (HADS) [33], the Dissociative Experiences Scale by Bernstein and Putnam 1982 [34], and the Resilience Scale by Connor and Davidson 2003 [35] that was administered twice in Group I (evaluations 1 and 4).

During each time point of evaluation, the frequency of SARS-CoV-2 infections (confirmed by reverse transcription-polymerase chain reaction or RT-PCR) was recorded, and participants replied to the following inventories (Section B.2): the short version of the Burnout Measure by Malach-Pines 2005 [36], the Depersonalization/Derealization Inventory by Cox and Swinson 2002 [37], the Pittsburgh Sleep Quality Index by Buysse 1989 [38], the short-form of the State-Trait Anxiety Inventory by Marteau and Bekker 1992 (STAIsv) [39], and the Stanford Acute Stress Questionnaire by Cardeña et al., 2000 [40].

After each of the five evaluations, a psychiatrist reviewed all the inventories, using the accepted criteria for each instrument; then, individualized reports were provided to the corresponding clinical department, with advice for counseling when required.

### 2.3. Analysis

Statistical analysis was performed using two-tailed *α* = 0.05 (STATISTICA software, StatSoft Inc, Tulsa, OK, USA). We assessed the data distribution using the Kolmogorov–Smirnov test. Since the inventory scores were not normally distributed, they are presented as medians and quartiles 1 and 3 (Q1–Q3). Comparisons between the groups of participants included in the study at the first or at the second epidemic waves (Group I vs. Group II), and comparisons by SARS-CoV-2 infection (infected participants vs. not infected), were performed using either the Mann–Whitney “*U*” test or the “*t*” test (for means or for proportions), according to the data distribution; comparisons within Group I were performed using Wilcoxon’s test. To assess linear correlations between inventory scores, we used Pearson’s correlation coefficient, and differences between coefficients were assessed by Fisher *r*-to-*Z* transformation. To assess score variations through time in each inventory, we used Friedman’s ANOVA. To assess the influence of cofactors (including SARS-CoV-2 infection) and inventory scores on changes in the report of acute stress through time, we used repeated measures multivariate analysis of covariance (MANCOVA). To assess linear and non-linear relationships among inventory scores and cofactors, we performed multivariable regression analyses, using a generalized linear model, with the Wald test and type 3 likelihood ratios.

## 3. Results

### 3.1. Characteristics of the Participants

Group I: This group was composed of 62 (56.8%) clinical workers, 13 (11.9%) technical and laboratory workers, and 12 (11%) support workers. A total of 14 (12.8%) reported smoking tobacco and 71 (65.1%) reported alcohol use; a total of 15 (13.7%) reported morbidities, including thyroid disorders (*n* = 6, 5.5%), high blood pressure (*n* = 4, 3.6%), and asthma (*n* = 3, 2.7%), and none of the participants reported psychiatric disorders.Group II: This group was composed of 24 (82.7%) clinical workers, 4 (13.7%) technical and laboratory workers, and 1 (3.4%) support worker. A total of 2 (6.8%) reported smoking tobacco and 21 (72.4%) reported alcohol use; two (8.8%) reported morbidities, which were thyroid disease (*n* = 1, 4.4%) and anxiety disorder (*n* = 1, 4.4%).

### 3.2. Comparisons According to the Time of Inclusion in the Study (Group I vs. Group II)

Compared to Group I, at the time of inclusion in the study, participants in Group II had higher scores on the depression subscale of the HADS, but similar scores on the total and partial anxiety scores of the HADS, the Dissociative Experiences Scale, and the Resilience Scale (Mann–Whitney “*U*” test, *Z* = 2.03, *p* = 0.042) (Table 1); they also had higher scores on the Pittsburgh Quality of Sleep Index (Mann–Whitney “*U*” test, *Z* = 2.06, *p* = 0.038) and the Stanford Acute Stress Questionnaire (Mann–Whitney “*U*” test, *Z* = 2.59, *p* = 0.009) (Table 2). Those who participated in the study since the beginning of the pandemic showed less depression, better quality of sleep, and less stress than those who had no psychological evaluations during the first epidemic wave.

### 3.3. Frequency of Infections by SARS-CoV-2 and Comparisons According to Infection

Among the participants in Group I, the frequency of infection by SARS-CoV-2 increased after each evaluation, up to a cumulative frequency of 44% at the fifth evaluation (Table 2). Compared to participants who were included in the study at the second epidemic wave (Group II), the cumulative frequency of infections was similar in the two groups (“*t*” test for proportions, *p* > 0.05) (Table 2). This result shows that the two groups had a similar exposure to the virus during the study period.

In all participants, at the second epidemic wave, comparisons according to the cumulative SARS-CoV-2 infections (i.e., infected vs. not infected) showed that those who were infected by the virus had the highest scores on the Burnout Measure (Mann–Whitney “*U*” test, *Z* = 2.02, *p* = 0.042), the Depersonalization/Derealization Inventory (Mann–Whitney “*U*” test, *Z* = 2.27, *p* = 0.022), the Pittsburgh Sleep Quality Index (Mann–Whitney “*U*” test, *Z* = 2.3, *p* = 0.019), and the Stanford Acute Stress Questionnaire (Mann–Whitney “*U*” test, *Z* = 2.07, *p* = 0.037) than those who were not infected by the virus (Table 3).

At the last evaluation, just before clinical departments were re-opened for patients with diseases other than COVID-19, and during COVID-19 vaccination for health workers, the majority of the differences observed at the second epidemic wave persisted (Mann–Whitney “*U*” test, *Z* > 2.00, *p* < 0.05), except for the Stanford Acute Stress Questionnaire (Table 3). These results may reflect the combined burden of the working conditions due to the sanitary emergency and that related to the disease and convalescence, in those who were infected, whereas COVID-19 vaccination could have decreased uncertainty and improved future expectations among all participants.

### 3.4. Simple Correlations among Inventory Scores

Linear correlations between the psychological screening inventories and the follow-up inventories were lower and less consistent for the Resilience Scale than for other scales. In contrast, as could be expected, the highest correlations and most consistent were those between the Dissociative Experiences Scale and the Depersonalization/Derealization Inventory, with decreasing values from the first evaluation (Pearson’s *r* = 0.80, *p* < 0.0001) to the last evaluation (*r* = 0.48, *p* < 0.0001) (*Z* = 4.17, *p* < 0.001), and those between the HADS and the STAIsv, with a similar decreasing pattern (Pearson’s *r* from 0.78, *p* < 0.0001, to 0.37, *p* < 0.0001) (*Z* = 4.78, *p* < 0.001). This result is consistent with changes in the report of psychological symptoms through time.

### 3.5. Changes through Time in Each Inventory Score

In Group I, the score on the Resilience Scale was similar before each of the two epidemic waves (Wilcoxon test, *p* > 0.05) (Table 1), and changes in each inventory score were as follows (Table 2):Burnout Measure: The lowest score was observed after the first epidemic wave and the highest at the peak of the second epidemic wave (Friedman’s ANOVA, *X*^2^ = 13.54, *p* = 0.008).Depersonalization/Derealization Inventory: The lowest score was observed after the first epidemic wave and the highest at the peak of the second epidemic wave (Friedman’s ANOVA, *X*^2^ = 14.89, *p* = 0.004).Pittsburgh Sleep Quality Index: The lowest score was observed after the second wave and the highest at the peak of the first epidemic wave (Friedman’s ANOVA, *X*^2^ = 21.22, *p* < 0.0003).STAIsv: The lowest score was observed before the peak of the first epidemic wave and the highest at the peak of the first epidemic wave (Friedman’s ANOVA, *X*^2^ = 26.91, *p* < 0.00002).Stanford Acute Stress Questionnaire: The lowest score was observed after the second wave and the highest at the peak of the first epidemic wave (Friedman’s ANOVA, *X*^2^ = 25.14, *p* < 0.00005).

The results show that symptoms of burnout and depersonalization/derealization increased through time. On the other hand, the highest reports of symptoms of acute stress and state anxiety, as well as bad quality of sleep, were evident during the first epidemic wave, when uncertainty prevailed. However, acute stress, state anxiety, and burnout were all related to trait anxiety/depression, with higher scores in participants with a HADS score ≥8 than in those with a HADS score <8 (Figure 2 and Figure 3).

In agreement with these results, during the first epidemic wave, the percentage of people reporting acute stress related to COVID-19 was 64.9%, while no stress was reported by 16.5% of participants, and stress related to personal and work issues was less frequent (circa 10% each).

However, during the second wave of infections, in Group I, the percentage of people reporting acute stress related to COVID-19 decreased to 43.1%, while 39.4% reported no stress, with almost no change in the percentage of people reporting acute stress related to personal or work issues (Figure 1). Consistently, at the second epidemic wave, in Group II, the percentage of people reporting acute stress related to COVID-19 was 51.7%, while no stress was reported by 13.7% of participants, and the percentage of people reporting acute stress related to either personal issues or work issues was 24.1% and 10.3%, respectively (Figure 4).

### 3.6. Acute Stress at Each Time Point of Evaluation

Figure 5 shows the frequency of acute stress reaction at each time point of evaluation. In Group I, the highest frequency was observed after the first epidemic wave, and it decreased during the second wave. However, in Group II, the highest frequency was at the second epidemic wave, with a significant difference between the two groups (“*t*” test for proportions, *t* = 2.07, *p* = 0.039).

During the follow-up, the scores on the Stanford Acute Stress Questionnaire were related to the following factors, with no evidence of influence from sex, or tobacco/alcohol use:In evaluation 1, before the peak of the first epidemic wave, the score on the Stanford Acute Stress Questionnaire was related to age (i.e., younger participants had higher scores than older participants), and to the scores on the Pittsburgh Sleep Quality Index, the Depersonalization/Derealization Inventory, and the Burnout Measure (Table 4) (Intercept Estimate 2.67 ± Standard Error 0.39, Wald Statistic 47.06, *p* < 0.0001).In evaluation 2, at the peak of the first epidemic wave, the Stanford Acute Stress Questionnaire was related to the scores on the Depersonalization/Derealization Inventory and the Burnout Measure (Table 4) (Intercept Estimate 1.42 ± 0.47, Wald Statistic 9.23, *p* = 0.002).In evaluation 3, after the first epidemic wave, the Stanford Acute Stress Questionnaire was related to the scores on the Pittsburgh Sleep Quality Index and the Depersonalization/Derealization Inventory (Table 4) (Intercept Estimate 2.07 ± 0.43, Wald Statistic 22.74, *p* < 0.0001).In evaluation 4, at the second epidemic wave, the Stanford Acute Stress Questionnaire was related to occupation, with an interaction between SARS-CoV-2 infections and occupation, and to the scores on the Pittsburgh Sleep Quality Index, the Depersonalization/Derealization Inventory, the STAIsv, and the Burnout Measure (Table 4) (Intercept Estimate 2.00 ± 0.44, Wald Statistic 19.93, *p* < 0.0001).In evaluation 5, after the second epidemic wave, the Stanford Acute Stress Questionnaire was related to age, the cumulative cases of SARS-CoV-2 infections, and occupation, with an interaction between SARS-CoV-2 infections and occupation, and to the scores on the Pittsburgh Sleep Quality Index, the Depersonalization/Derealization Inventory, and the Burnout Measure (Table 4) (Intercept Estimate 2.41 ± 0.47, Wald Statistic 26.36, *p* < 0.0001).

During the follow-up, higher estimates were observed for the relationship between burnout and acute stress, particularly just before the first epidemic wave and after the second wave. At the same time points, younger age was related to higher scores. Although the influence of occupation was not evident during the surge in patients at the two epidemic waves, it was evident at the end of the follow-up. Technicians reported more acute stress than clinical workers. In addition, at the last evaluation, workers who were infected by SARS-CoV-2 reported more acute stress than those who were not infected, with an interaction with occupation.

## 4. Discussion

During the follow-up of frontline health workers in two consecutive epidemic waves of COVID-19, the maximum reports of acute stress (related to COVID-19), state anxiety, and bad quality of sleep were observed when uncertainty prevailed, at the first surge in patients, while burnout increased through time, and resilience was invariant. After the first epidemic wave, acute stress decreased, with no increase during the second wave, and further decreased when vaccination started. However, acute stress, state anxiety, and burnout were all related to trait anxiety/depression, with higher scores in participants with a HADS score ≥8 than in those with a HADS score <8 (Figure 2 and Figure 3), which may allow early identification of workers who are prone to acute stress. Although the influence of occupation was not evident during the surge in patients, at the end of the follow-up, technicians reported more acute stress than clinical workers, with an interaction with the cumulative SARS-CoV-2 infection. In addition, compared to participants included in the study at the beginning of the pandemic, participants included in the study at the second epidemic wave reported more symptoms of depression and bad quality of sleep, with a higher frequency of acute stress reaction during the second epidemic wave.

These results are consistent with a repeated cross-sectional study of the mental health status of intensivists at the first two epidemic waves in Italy that showed increased anxiety and bad sleep during the first wave, while compassion fatigue and depression increased at the second wave [41]. The findings are also consistent with variation in occupational stressors as the pandemic evolved. Internationally, at the beginning of the sanitary emergency, concerns regarding preparation and resources prevailed; the fear of infection, safety measures, and general uncertainty were the main stressors [42]. However, the prolongation of the epidemic resulted in persistent, increased workload levels with isolation, and frequent negative outcomes for the patients that had an impact on the mental health of those at the forefront of healthcare; stress was associated with anxiety, depression, burnout, and dissatisfaction [41]. Contrariwise, after the second epidemic wave, vaccination may have reduced anxiety [43] while improving expectations for pandemic control [41]. However, more evidence is required regarding the factors related to the variety of occupations encompassing the healthcare of patients with COVID-19. In this study, the small number of non-clinical participants did not allow us to appropriately assess dissimilar responses according to occupation.

During this study, at the beginning of the pandemic, trait and state anxiety allowed for identifying the participants who were more susceptible to reporting acute stress. Trait anxiety is considered a dispositional characteristic of an individual's personality to respond to a potential threat, while state anxiety is an acute response to a specific threat (for a review, see [44]). Trait and state anxiety are not mutually exclusive, and state anxiety triggered by an event can be superimposed on trait anxiety [45]; however, the situation must be congruent with the dimension of trait anxiety in order to evoke increases in state anxiety [44]. Moreover, unconditioned anxiety can be enhanced by prior stress [46], while acute stress may induce a state anxiety response, which facilitates increased attention, sensory processing, and executive functions mainly mediated by noradrenaline release (for a review, see [47]).

An effect of age was observed both at the start and at the end of the follow-up. Animal studies support the idea that stress effects on the central nervous system vary as a function of age, with age-specific patterns of neuronal activation and neuroendocrine responses [48]. In humans, compared to young adults, older adults employ more adaptive and directive approaches to managing distress [49,50]; we did not observe differences related to sex. These results are in agreement with the findings of a cross-sectional survey of 1452 health workers, in Spain, showing that age group (30–39 years old), being a nurse or nursing assistant, and symptoms of anxiety/depression are variables that independently increase the probability of requiring psychological assistance, with no influence from sex [51].

Previous cross-sectional studies have shown that approximately 5% of health workers caring for patients with COVID-19 may suffer severe acute stress [16,52,53]. In addition, we observed that the progress of the COVID-19 pandemic compelled health workers to cope with consecutive challenges, within similar working conditions. In this study, the majority of participants were able to adjust and deal with the stress during the second epidemic wave, as evidenced by the acute stress decrease, even if state anxiety persisted, and burnout increased. This result is consistent with the assumption that, in aversive environments, unpredictable threats or stressors [54] may enable either avoidance or appropriate responses to meet future threats that become somehow predictable (for a review, see [55]). We also observed the expected strong correlation between depersonalization/derealization symptoms and acute stress [56,57]. However, further studies are needed to assess the consequences of burnout and dissociation in health workers on the attention to the human aspects of severe illness [58].

We observed that during the second epidemic wave, compared to participants who were evaluated during the two epidemic waves, those with no previous follow-up showed the highest frequency of acute stress reaction during the second epidemic wave; however, repeated exposure to a similar stressor may produce either habituation or sensitization, resulting in a reduced/increased response to subsequent events [59]. These participants also had higher scores on depression symptoms and poor quality of sleep. This is coherent with the evidence that, in healthy adults, anxiety and depression symptoms are associated with stress reactivity and recovery by either blunted or exaggerated cortisol responses to and recovery from stress [4]. It is also consistent with studies supporting the influence of chronic stress and bad sleep on affective health and acute stress experience [60]. Nevertheless, we have to consider that, after each evaluation, we reported the psychological symptoms of the participants to their corresponding clinical department. Thus, the follow-up of the participants in Group I during the exposure to the stressors might have influenced their responses during subsequent evaluations.

During the follow-up, changes in sleep quality and acute stress showed a similar pattern, where the highest score occurred during the first epidemic wave, and the lowest after the second epidemic wave. This finding is consistent with a report of a high prevalence of both poor sleep quality and moderate-severe stress in healthcare workers (either at the frontline or not), during the COVID-19 pandemic [32,61]. However, the effects between sleep and stress are bidirectional [62,63]; hyperactivity of the hypothalamic–hypophyseal–adrenal axis can have negative effects on sleep, while sleep disturbances can exacerbate the dysfunction of this axis (for a review, see [63]). In addition, repeated or chronic activation of the hypothalamic–hypophyseal–adrenal axis may bring about dysfunction [3]. Furthermore, evidence suggests relationships among dissociation, insomnia, and distress tolerance [64,65]. The results highlight the need for long-term attention to sleep quality and stress in health workers, further considering that both stressors and elevated burnout may persist during and long after the pandemic.

The results support the notion that timely psychological evaluation of health workers may be valuable to implement strategies to prevent or moderate adverse psychological reactions caused by sanitary crises. This study is consistent with the increasing evidence calling for long-term mental health protection and promotion in health workers, including active participation of mental health professionals at all levels, from policy design to clinical practice [51,66]. The World Psychiatric Association position paper has stressed that [66]: “While variations across countries will exist in responding to the COVID-19 pandemic, the human rights of individuals with mental disorders must be protected, and appropriate and safe services provided for their treatment. Moreover, the negative impact of the pandemic on government budgets should not be used as an excuse to reduce essential services for people with mental illness during or after the pandemic. Psychiatrists can play important roles in advocating for these measures and in supporting their patients, colleagues and the healthcare system's response to the pandemic”.

The main limitation of this study is the reliance on self-report; yet, direct assessment was performed within the working environment of each participant. The second limitation is the lack of a reference group during the complete follow-up; however, a second group of participants was assessed just during the second epidemic wave. In addition, SARS-CoV-2 infection among participants implied that responses may reflect the combined burden of the sanitary emergency and that related to the disease/convalescence. Conversely, the main strength of this study is the follow-up during an extraordinary natural disaster, which allowed recording the reactions to consecutive similar stressors, within the same working conditions.

Since the ability to adapt stress responses to changing environments depends on individual factors and previous experience [3,55], while genetic influences on brain function may be modulated by the environmental context [67], attempts to mitigate acute stress among frontline health workers during a sanitary emergency should take into account both individual factors (traits and experience) and stressors related to the context.

## 5. Conclusions

In health workers exposed to stressors related to repeated epidemic waves of infections, early assessment of trait and state anxiety/depression and quality of sleep (with counseling) could be useful to identify individual traits and enable strategies for coping with successive sanitary crises. Overall, the results emphasize the benefits of mental health screening in order to identify mental health problems that could be treated before, during, and after an epidemic wave of infections.

## Figures and Tables

**Figure 1 ijerph-19-00206-f001:**
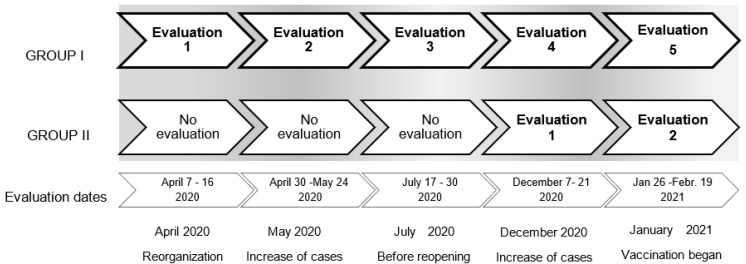
Time points of evaluation during follow-up of the participants included either at the first epidemic wave (Group I) or the second epidemic wave (Group II).

**Figure 2 ijerph-19-00206-f002:**
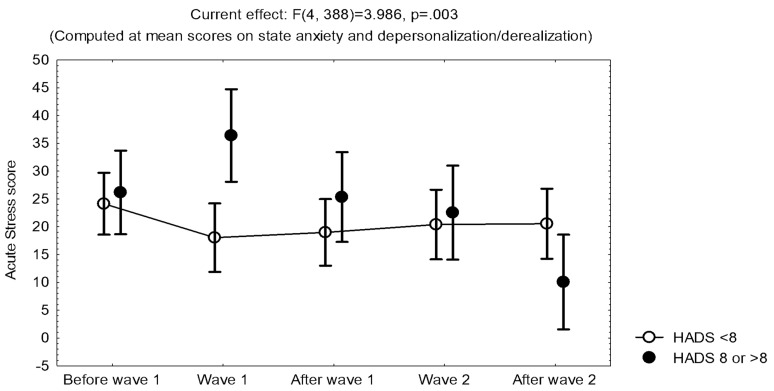
Mean and standard error of the mean of the scores on the Stanford Acute Stress Questionnaire during the 5 evaluations in Group I (*n* = 109), according to the score on the Hospital Anxiety and Depression Scale (HADS) (< 8 or ≥8).

**Figure 3 ijerph-19-00206-f003:**
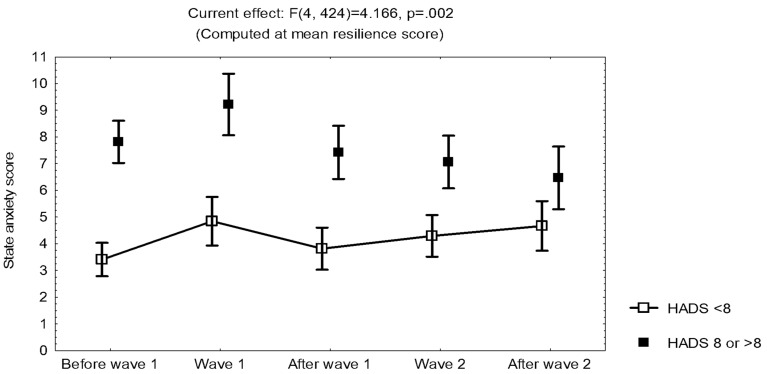
Mean and standard error of the mean of the scores on the State-Trait Anxiety Inventory (short version) during the evaluations in Group I (*n* = 109), according to the score on the Hospital Anxiety and Depression Scale (HADS) (<8 or ≥8).

**Figure 4 ijerph-19-00206-f004:**
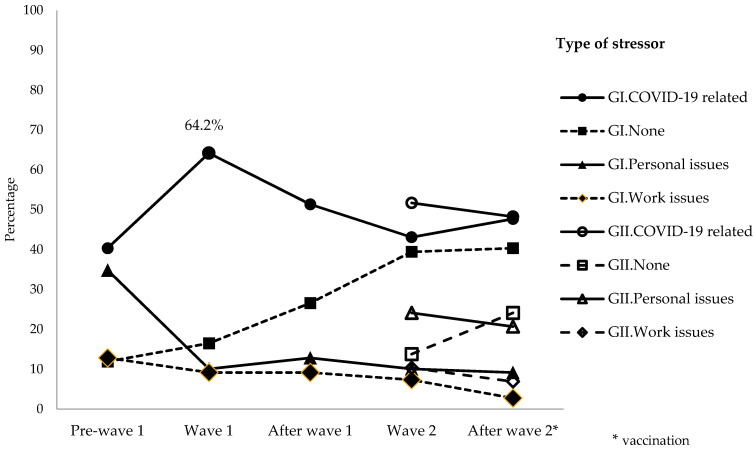
Frequency of reporting acute stress, according to the type of stressor, during the follow-up of participants included in the study, either at the first epidemic wave (GI) or the second epidemic wave (GII). The time of vaccination is highlighted with an asterisk.

**Figure 5 ijerph-19-00206-f005:**
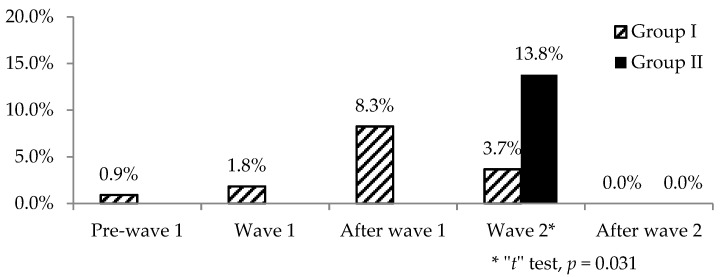
Frequency of acute stress reaction during the 5 evaluations performed in Group I and the 2 evaluations performed in Group II. A significant difference is highlighted with an asterisk.

**Table 1 ijerph-19-00206-t001:** Median and quartiles 1 and 3 (Q1–Q3) of the total score of the Resilience Scale, the Hospital Anxiety and Depression Scale (HADS), and the Dissociative Experiences Scale at the time of inclusion in the study, either at the first epidemic wave (Group I) or at the second epidemic wave (Group II).

Inventories	Group I Evaluation 1 (*n* = 109)	Group II Evaluation 4 (*n* = 29)	*p* < 0.05
Resilience Scale	median (Q1–Q3)	median (Q1–Q3)	
Pre-wave 1	80 (69–85)		
Wave 2	78 (66–86)	81 (67–86)	-
Hospital Anxiety and Depression Scale			
Depression score	2 (1–5)	4 (2–6)	0.042
Anxiety score	6 (3–10)	6 (4–11)	-
Total score	8 (4–16)	11 (5–16)	-
Dissociative Experiences Scale	5.7 (3.2–11.7)	6 (2.5–10)	-

**Table 2 ijerph-19-00206-t002:** Frequency of infection by SARS-CoV-2, and median with quartiles 1 and 3 (Q1–Q3) of the inventory scores, during the follow-up of 138 frontline health workers, in the course of the first and second epidemic waves, according to the time point at inclusion in the study. Comparisons were performed using the “*t*” test for proportions and the Mann–Whitney “*U*” test.

	Pre-Wave 1		Wave 1		After Wave 1		Wave 2			After Wave 2		
	Group I (*n* = 109)	II	Group I (*n* = 109)	II	Group I (*n* = 109)	II	Group I (*n* = 109)	Group II (*n* = 29)	*p* < 0.05	Group I (*n* = 109)	Group II (*n* = 29)	*p* < 0.05
SARS-CoV-2 infection												
Frequency of Infection (*n*)	0.09% (1)	-	8.2% (9)	-	16.5% (18)	-	41.2% (45)	41.3% (12)	-	44.0% (48)	48.2%(14)	-
Inventories (median, Q1–Q3)												
Depersonalization/Derealization	5 (1–12)	-	3 (1–10)	-	4 (0–12)	-	5 (13)	9 (3–16)	-	3 (0–8)	5 (2–11)	-
Pittsburgh Sleep Quality Index	8 (5–10)	-	8 (5–11)	-	8 (5–10)	-	5 (2–8)	8 (6–11)	0.038	6 (4–9)	8 (5–10)	-
Stanford Acute Stress Questionnaire	17 (4–37)	-	10 (1–40)	-	10 (0–34)	-	4 (0–32)	17 (12–44)	0.009	3 (0–21)	17 (2–29)	-
State-Trait Anxiety Inventory (s.v.)	5 (3–7)	-	6 (3–9)	-	5 (2–7)	-	5 (2–8)	6 (4–8)	-	5 (3–8)	5 (1–7)	-
Burnout Measure (short-form)	2.2 (1.5–2.7)	-	2 (1.5–3.2)	-	2 (1.4–2.9)	-	2.2 (1.6–3)	2.5 (1.8–3.1)	-	1.9 (1.4–2.7)	2.3 (1.7–3.1)	-

**Table 3 ijerph-19-00206-t003:** Median and quartiles 1 and 3 (Q1–Q3) of the inventory scores of 138 participants during the second epidemic wave, according to SARS-CoV-2 infection. Comparisons were performed using the Mann–Whitney “*U*” test.

	Wave 2			After Wave 2		
Inventories	Infection (*n* = 57)	No Infection (*n* = 81)	*p* < 0.05	Infection (*n* = 62)	No Infection (*n* = 76)	*p* < 0.05
	Median (Q1–Q3)	Median (Q1–Q3)		Median (Q1–Q3)	Median (Q1–Q3)	
Depersonalization/Derealization	7 (2–21)	3 (0–12)	0.022	5 (1–12)	2 (0–7)	0.036
Pittsburgh Sleep Quality Index	9 (6.5–12)	6 (4–10)	0.019	8 (5–10)	5 (4–8.5)	0.045
Stanford Acute Stress Questionnaire	16 (0–44)	3 (0–28)	0.037	6 (3–9)	5 (3–7)	-
State-Trait Anxiety Inventory (s.v.)	5 (3–9)	6 (2–8)	-	6 (3–9)	5 (3–7)	-
Burnout Measure (short-form)	2.7 (1.7–3.6)	2 (1.6–2.9)	0.042	2.3 (1.6–3.1)	1.8 (1.3–2.7)	0.020

**Table 4 ijerph-19-00206-t004:** Results of the multivariable analysis on the score of the Stanford Acute Stress Questionnaire at each time point of evaluation, including the follow-up inventories, age, occupation, and cumulative SARS-CoV-2 infection. Statistical significance is highlighted in bold.

	Pre-Wave 1 Estimate ± S.E.	Wave 1 Estimate ± S.E.	After Wave 1 Estimate ± S.E.	Wave 2 Estimate ± S.E.	After Wave 2 Estimate ± S.E.
Inventories					
Pittsburgh Sleep Quality Index	0.090 ± 0.019	0.038 ± 0.024	0.118 ± 0.029	0.050 ± 0.022	0.048 ± 0.019
*X*^2^ (*p* value)	19.37 (**<0.0001**)	2.23 (0.134)	14.46 (**0.0001**)	4.60 (**0.031**)	7.23 (**0.007**)
State-Trait Anxiety Inventory (s.v.)	0.006 ± 0.017	0.021 ± 0.017	0.034 ± 0.022	0.087 ± 0.021	0.080 ± 0.031
*X*^2^ (*p* value)	0.09 (0.764)	1.19 (0.274)	1.73 (0.187)	18.96 (<**0.0001**)	8.73 (**0.003**)
Depersonalization/Derealization	0.011 ± 0.004	0.025 ± 0.006	0.022 ± 0.006	0.013 ± 0.004	0.011 ± 0.005
*X*^2^ (*p* value)	4.65 (**0.030**)	18.71 (<**0.0001**)	9.66 (**0.001**)	17.42 (<**0.0001**)	4.42 (**0.035**)
Burnout Measure	0.248 ± 0.069	0.026 ± 0.010	−0.007 ± 0.009	0.135 ± 0.059	0.253 ± 0.090
*X*^2^ (*p* value)	9.69 (**0.001**)	5.37 (**0.020**)	0.64 (0.423)	4.76 (**0.029**)	9.21 (**0.002**)
Cofactors					
Age	−0.029 ± 0.010	0.014 ± 0.011	−0.005 ± 0.012	−0.013 ± 0.011	−0.039 ± 0.010
*X*^2^ (*p* value)	8.70 (**0.003**)	1.16 (0.280)	0.15 (0.691)	1.21 (0.270)	16.01 (**0.0001**)
Occupation (clinical vs. technical)	−0.186 ±0.098	−0.129 ±0.148	0.136 ± 0.877	0.109 ± 4.329	−0.765 ± 0.128
*X*^2^ (*p* value)	3.42 (0.180)	3.17 (0.205)	1.34 (0.510)	4.86 (0.087)	33.97 (**<0.0001**)
SARS-CoV-2 infection	-	−0.370 ± 0.138	−0.143 ± 0.095	−0.012 ± 0.076	0.174 ± 0.077
*X*^2^ (*p* value)	-	1.65 (0.198)	1.61 (0.203)	0.02 (0.864)	5.38 (**0.020**)
Occupation * SARS-CoV-2 infection		−0.129 ± 0.148	−0.039 ± 0.116	−0.142 ± 0.089	−0.329 ± 0.093
*X*^2^ (*p* value)		3.17 (0.205)	0.51 (0.773)	7.20 (**0.027**)	16.0 (**0.0003**)

## Data Availability

Data are available at: https://dataverse.harvard.edu/dataset.xhtml?persistentId=doi:10.7910/DVN/3RAES2 (accessed on 11 November 2021).

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
