# Peer review of "Acute Stress in Health Workers during Two Consecutive Epidemic Waves of COVID-19"

_ijerph, 2021, doi:10.3390/ijerph19010206_

Round 1

Reviewer 1 Report

Dear Authors, this is great work:  I work as Director of Anaesthesia and Intensive Care Department: I’ve found this research very interesting.

The introduction is essential and precise

Methods are adequately described

Results are evident and interesting

Limits are well discussed

Discussion:

I think it will be interesting to develop and better discuss the reasons for stress related to COVID-19 in health care workers. In many cases, physicians have played a crucial role in managing grief and sometimes accompanying patients in the final phase of their life. I recommend this paper (How to humanize the intensive care unit BMJ ebm, I’m not the author, and you are not obliged)

Then in our experience, our medical equipe also had a lot of physical diseases after the covid-19 pandemic. Do you have some data? I think you should integrate this topic. Also, the workload stress could have promoted this process?

Overall comment: the research is exciting and well-conducted: it deserves to be published after minor revision

Author Response

I think it will be interesting to develop and better discuss the reasons for stress related to COVID-19 in health care workers. In many cases, physicians have played a crucial role in managing grief and sometimes accompanying patients in the final phase of their life. I recommend this paper (How to humanize the intensive care unit BMJ ebm, I’m not the author, and you are not obliged)

- We thank the reviewer for the comments and for the reference, which has been included in the discussion (reference 52).

- We have also included the main stressors (second paragraph of the Discussion section).

Then in our experience, our medical equipe also had a lot of physical diseases after the covid-19 pandemic. Do you have some data? I think you should integrate this topic. Also, the workload stress could have promoted this process?

- The participants section now includes the report of comorbidities at the time of  inclusion in the study (section 2.1). The low rate of illness among participants is supportive of the results. However, during the follow-up we didn’t record the general health.

Author Response

Format

  1. There are two”3.4” in the article. The latter one should be “3.5”.

- Thank you for noticing the mistake, it has been corrected.

  1. “p” and “t” in the results part and tables should be italic.

- The font style has been modified accordingly.

  1. Different icons in figure 2 are difficult to distinguish. It is suggested to change icons.

- The icons have been modified.

  1. The typography could be improved, especially the results section.

- Thank you, the typography has been revised.   

Content.

  1. A description of the relationship between the selected psychological variables and the stressful event is recommended in the introduction section.

- We thank you for the comment, the Introduction has been edited accordingly.

  1. The difference between the number of subjects in two groups and the selection criteria needs further explanation, while the description of subjects who withdrew from the study can be appropriately streamlined.

-The participants section was edited to state that no sample size was calculated and that the invitation to participate was open for frontline health workers holding the same position during the two epidemic waves. The sample size of each group was given by real world circumstances, since the majority of the front-line workers in the hospital participated during the first study about posttraumatic stress disorder (Group I). Afterwards, we became aware of a missing subgroup, and invited them to participate for this second study (after institutional authorization) (Group II).  

-The streamline is now shown in Appendix 1.

  1. In the Procedure, it is suggested the time points of evaluation are tabulated as wellas the graphs could be improved to be clearer and more concise.

- Tabulation was performed and all the Figures have been edited for clarity.

  1. In the results section, effect size should be reported.

The effect size can refer to a standardized measure such as r or Cohen's d, or to unstandardized measures such as unstandardized regression coefficients. However, effect size can be generalized by using multivariate measures of association, such as the analysis that we performed.

Though, the standardized mean difference (or Cohen’s d) estimates the effect size between two means, appropriate calculation requires considering data distribution and sample size. In this study, comparisons were on data with a distribution different than normal and unequal sample size groups. Then, we provide the effect size of the main results by describing the correlation coefficients and the Estimates (with their standard errors) through the text and Tables.

  1. The following variables might may affect the results which haven’t been controlled: (1)sex ratio is inconsistent between two groups.(2)subjects’ original anxiety and related levels are not controlled.

- The results of the analysis showed no contribution of sex, which is described in the results section “During follow-up, the scores on the Stanford Acute Stress Questionnaire were related to the following factors, with no evidence of influence from sex, or tobacco/ alcohol use…” (section 3.5) and the Discussion.

- The results section was edited to highlight the influence of trait anxiety (“original” anxiety”), and figures 4 and 5 (showing the influence of trait anxiety) became figures 2 and 3 (section 3.3). This result is further discussed in the corresponding section.

  1. In general, it is difficult to correspond between the main body of the article and the conclusion. Please provide further explanation of the value of the application of mental health screening in the discussion section.

- The Discussion has been edited accordingly.

Round 2

Reviewer 2 Report

I would recommend to accept this manuscript.

Author Response

Thank you for the review to improve the manuscript.